# Regulatory Mechanisms of Yili Horses During an 80 km Race Based on Transcriptomics and Metabolomics Analyses

**DOI:** 10.3390/ijms26062426

**Published:** 2025-03-08

**Authors:** Jianwen Wang, Wanlu Ren, Zexu Li, Luling Li, Ran Wang, Shikun Ma, Yaqi Zeng, Jun Meng, Xinkui Yao

**Affiliations:** 1College of Animal Science, Xinjiang Agricultural University, Urumqi 830052, China; wjw1262022@126.com (J.W.); 13201295117@163.com (W.R.); 13593312012@163.com (Z.L.); 18996888638@163.com (L.L.); 17590811761@163.com (R.W.); 18299152719@163.com (S.M.); xjauzengyaqi@163.com (Y.Z.); 2Xinjiang Key Laboratory of Equine Breeding and Exercise Physiology, Urumqi 830052, China

**Keywords:** Yili horse, endurance racing, transcriptome, metabolome, regulatory mechanism

## Abstract

Equine endurance exercise induces physiological changes that alter metabolism and molecular pathways to maintain balance after intense physical activity. However, the specific regulatory mechanisms remain under debate. Identifying differentially expressed genes (DEGs) and differential metabolites (DMs) associated with equine endurance is essential for elucidating these regulatory mechanisms. This study collected blood samples from six Yili horses before and after an 80 km race and conducted transcriptomics and metabolomics analyses, yielding 722 DEGs and 256 DMs. These DEGs were primarily enriched in pathways related to amino acid biosynthesis, cellular senescence, and lipid metabolism/atherosclerosis. The DMs were predominantly enriched in fatty acid biosynthesis and the biosynthesis of unsaturated fatty acids. The integrative transcriptomics and metabolomics analyses of DEGs and DMs highlight functional changes during the endurance race. The findings offer a holistic understanding of the regulatory mechanisms underlying equine endurance and a solid foundation for formulating training programs to optimize horse performance in endurance racing.

## 1. Introduction

Endurance racing is a prominent equestrian sport. According to the Fédération Equestre Internationale (FEI) regulations, the typical race distance ranges from 80 to 160 km [1]. This imposes significant demands on the physiological adaptability of horses. Transcriptomics and metabolomics, as integral parts of systems biology, have emerged as frontier fields in exploring equine endurance performance. During endurance racing, the molecular mechanisms [2,3] and physiological adaptations [4] of horses under sustained exercise conditions play a crucial role in their performance. Therefore, delving into the relationship between transcriptomics and metabolomics in endurance racehorses can provide scientific insights for enhancing horse performance and refining training strategies.

Transcriptomics primarily involves the study of intracellular RNA expression, thoroughly analyzing gene expression profiles in tissues and cell types under specific conditions. Gene expression in horses is notably influenced by varying training intensities [5,6,7]. It has been reported that endurance racing can lead to substantial upregulation of specific genes related to energy metabolism, oxygen transport, and anti-oxidative stress [8,9]. Additionally, endurance training can enhance the expression of genes related to mitochondrial biogenesis in muscles, such as *PPAR* γ (Peroxisome proliferator-activated receptor gamma), FRX-CREB axis, and their downstream targets. The upregulation of these genes can strengthen mitochondrial function, thereby improving muscle endurance [10]. Consequently, changes in gene expression directly impact the adaptability and performance enhancement of horses in prolonged exercise [11,12].

Metabolomics examines dynamic changes in small molecules within cells or organisms, favoring in-depth comprehension of the physiological mechanisms of endurance racehorses. During such racing, horses primarily consume energy from the breakdown of glycogen, fats, and proteins [13,14]. Metabolomics analysis can elucidate metabolic pathways and dynamic shifts associated with these processes. Studies have disclosed that endurance racing induces variations in the concentrations of primary energy metabolites, including lactate [15], creatine [16], and amino acids [17]. Lactate, as a major product of anaerobic metabolism, indicates a shift in energy metabolism when its levels increase [15]. Moreover, changes in certain metabolites, such as free fatty acids [18] and ketone bodies [19], can reflect the switching of energy sources under diverse exercise intensities. Metabolomics analysis in endurance training and racing can also assess antioxidant capacity and inflammatory responses [20], which are closely tied to horses’ overall physiological state.

The physiological adaptability of endurance racehorses is complicatedly regulated by transcriptome and metabolome. For instance, the metabolite AMPK (adenosine monophosphate-activated protein kinase) can activate transcription factors associated with energy metabolism and impact fatty acid oxidation and glucose metabolism, playing a pivotal role in endurance exercise [21,22]. Additionally, reactive oxygen species (ROS) generated during exercise function in signal transduction at certain concentrations, modulating the expression of oxidative stress-related genes and enhancing cellular antioxidant capacity [23]. This process is vital for the adaptation of horses to endurance racing.

This study performed transcriptomics and metabolomics analyses of blood samples from Yili horses before and after an 80 km race, examining changes in their blood transcriptome and metabolome. It provides a deeper understanding of the physiological mechanisms of endurance racehorses and offers theoretical support for improving their performance and refining training and management strategies.

## 2. Results

### 2.1. Transcriptome Sequencing Quality Control

The transcriptomics analysis of 12 samples yielded 83.06 Gb of sequence data. The T0 and T1 groups generated 274,670,154 and 279,090,448 reads, respectively. Through quality control, 270,828,332 and 275,090,926 clean reads were retained, with each sample having clean data exceeding 6.15 Gb. The Q30 base percentage for all results was 94.10% or higher. Alignment of clean reads with the horse reference genome exhibited data validity rates between 90.23% and 95.49%, indicating high quality and completeness of the transcriptome data, suitable for subsequent analysis (Table 1).

### 2.2. DEG Screening and Expression Analysis

Using |log2(FoldChange)| ≥ 2 and padj ≤ 0.01 as the criteria, 722 DEGs were identified between the two groups. Compared to the T0 group, the T1 group had 471 DEGs significantly upregulated and 251 ones significantly downregulated. The volcano plot is shown in Figure 1. Through GO functional enrichment analysis of these DEGs, the top 10 GO terms for biological process (BP), cellular component (CC), and molecular function (MF) were selected, respectively, plotting a scatter diagram (Figure 2). Among them, eight GO terms were significantly enriched, all belonging to MF: transferase activity, transferring phosphorus-containing groups (GO:0016772), nucleoside binding (GO:0001882), GTP binding (GO:0005525), guanyl nucleotide binding (GO:0019001), kinase activity (GO:0016301), protein serine/threonine kinase activity (GO:0004674), GTPase activity (GO:0003924), and zinc ion binding (GO:0008270). KEGG enrichment analysis was conducted on the identified DEGs, and a scatter plot of the top 20 pathways was illustrated (Figure 3). Among these pathways, 17 were significantly enriched, including NOD-like receptor signaling pathway (ecb04621), chemokine signaling pathway (ecb04062), osteoclast differentiation (ecb04380), Kaposi sarcoma-associated herpesvirus infection (ecb05167), salmonella infection (ecb05132), leukocyte transendothelial migration (ecb04670), biosynthesis of amino acids (ecb01230), Th17 cell differentiation (ecb04659), endocytosis (ecb04144), bacterial invasion of epithelial cells (ecb05100), PD-L1 expression and PD-1 checkpoint pathway in cancer (ecb05235), pertussis (ecb05133), cellular senescence (ecb04218), lipid metabolism/atherosclerosis (ecb05417), Fc gamma R-mediated phagocytosis (ecb04666), DNA replication (ecb03030), and B cell receptor signaling pathway (ecb04662).

### 2.3. Real-Time Fluorescence Quantitative PCR Validation

Four genes were randomly selected for real-time fluorescence quantitative PCR validation. The results are shown in Figure 4. The quantitative changes in the genes agree with the expression trend of transcriptome sequencing results, and the expression levels of four genes in T1 group were higher than those in T0 group, denoting the reliability of transcriptomics analysis.

### 2.4. DM Screening and Analysis

PCA of six samples per time point (T0 and T1) revealed a clear separation between the two groups (Figure 5). Using VIP > 1.0, FC > 2/FC < 0.5, and *p*-value < 0.05 as criteria, a total of 256 DMs were identified. Compared to the T1 group, the T0 group had 20 significantly upregulated and 236 significantly downregulated DMs. The volcano plot of DMs is shown in Figure 6. KEGG functional annotation and pathway enrichment analysis were performed to further explore the biological functions of DMs. A bubble chart (Figure 7) was plotted for the top 10 pathways with the smallest *p*-values. With *p*-value < 0.1 as the threshold, four significantly enriched pathways were identified: fatty acid biosynthesis (map00061), biosynthesis of unsaturated fatty acids (map01040), arachidonic acid metabolism (map00590), and one carbon pool by folate (map00670). The involved DMs encompass myristic acid, palmitoleic acid, palmitic acid, lauric acid, stearic acid, adrenic acid, arachidic acid, docosahexaenoic acid, docosapentaenoic acid, linoleic acid, 5-Oxo-ETE, prostaglandin g2, 16(R)-HETE, oleic acid, and folinic acid.

### 2.5. Correlation Analysis

All identified DEGs and DMs were mapped to the KEGG pathway database to obtain their common pathway information, identifying their co-involved major biochemical and signaling pathways. The shared pathways include fatty acid biosynthesis, carbon metabolism, biosynthesis of amino acids, and platelet activation. These pathways are linked to nine DMs (Myristic Acid, 2-Isopropylmalic Acid, Prostaglandin H2, Palmitoleic Acid, Palmitic Acid, Lauric Acid, Stearic Acid, Gluconolactone, and L-cysteine) and 19 DEGs (*GLUL* (Glutamate-Ammonia Ligase), *ADCY4* (Adenylate Cyclase 4), *LOC100051292*, *TKT* (*Transketolase*), *VASP* (Vasodilator Stimulated Phosphoprotein), *LOC100630899*, *LOC102147826*, *GAPDH* (Glyceraldehyde-3-phosphate dehydrogenase), *FCER1G* (Fc Epsilon Receptor Ig), *LOC106782187*, *LOC100054877*, *ACSL4* (Acyl-CoA Synthetase Long Chain Family Member 4), *LOC100147417*, *PC* (Pyruvate Carboxylase), *LOC100050395*, *MAPK11* (Mitogen-Activated Protein Kinase 11), *ALDOC* (Aldolase, Fructose-Bisphosphate C), *MAPK12* (Mitogen-Activated Protein Kinase 12), and *HSD17B8* (Hydroxysteroid 17-Beta Dehydrogenase 8)).

Pearson correlation analysis was performed on these DEGs and DMs. The correlation heatmap was generated (Figure 8). In the heatmap, blue and red indicate negative and positive correlations, respectively. The color depth is positively related to the correlation degree (deeper red implies a stronger positive correlation, and deeper blue signifies a stronger negative correlation). The more elongated the ellipses, the higher the absolute values of correlation.

## 3. Discussion

Prior studies of Thoroughbreds have identified 91 DEGs in blood and muscle samples before and after exercise [24]. Similarly, numerous DEGs in Arabians contribute to maintaining homeostasis during exercise [25]. This study screened 722 DEGs, aligning with the findings in Thoroughbreds. This suggests that these DEGs may regulate exercise-triggered signaling pathways. Additionally, DEGs were significantly enriched in GO terms related to nucleotide, GTP, zinc ion binding, and kinase activity, implying that DEG-encoded proteins act as regulators or components of kinase activity. These DEGs are potentially involved in exercise responses and sustain homeostasis by modulating specific physiological pathways [5].

The DEGs in this study were notably enriched in the biosynthesis of amino acids. During endurance competitions, muscles need substantial energy. Given the limitations of glycogen stores, alternative energy sources that rely on endogenous biosynthesis are crucial [26]. It has been reported that pathways such as Glycine, serine, and threonine metabolism can ameliorate maximum muscle power output [27]. This is because amino acids like arginine and glycine in the liver synthesize creatine, which is then converted into phosphocreatine to fuel muscle activity [28]. Furthermore, studies on high-level endurance athletes have demonstrated that γ-glutamyl groups transferred from glutathione to amino acids preserve intracellular homeostasis under oxidative stress [29]. Therefore, amino acid biosynthesis is vital for sustaining high-level energy output in muscles over extended periods. In this pathway, *GLUL* is significantly enriched. The proteins encoded by *GLUL* are involved in glutamine synthesis. High GLUL activity generally facilitates the biosynthesis of glutamine [30], an amino acid essential for protein and energy metabolism. During post-endurance recovery, glutamine supplementation is often insufficient [31]. The findings in this study disclose a significant upregulation in *GLUL* expression in blood post-exercise, representing heightened glutamine biosynthesis capacity in horses after prolonged endurance training, which can promote their capability to provide sustained energy during intense exercise. Additionally, GLUL can inhibit macrophage polarization to M1 [32], thereby adjusting immune function. It has been proven that endurance exercise can raise inflammation markers in the blood [33,34]. In this study, the evident rise in GLUL post-exercise strengthens immune capacity, potentially benefiting endurance performance.

Cellular senescence is characterized by the synthesis of distinctive pro-inflammatory secretome and protease, reduced autophagy, and cell growth arrest [35]. Prolonged exercise may generate ROS, leading to telomere shortening and DNA damage [36]. Meanwhile, cellular senescence may be associated with telomere dysfunction and DNA damage related to growth factor signaling [37]. This might explain the enrichment of the cellular senescence pathway in this study. Existing research has revealed that athletes have longer telomeres than non-athletes [36], suggesting that prolonged high-level exercise may assist in retarding cellular senescence [38]. In this transcriptome study, *GADD45A* (Growth Arrest and DNA Damage) in the cellular senescence pathway was significantly upregulated post-exercise. This gene is involved in DNA repair, genomic stability, and cell cycle arrest [39]. *GADD45A*-deficient mice exhibit declined DNA repair and severe genomic instability [40]. The upregulation of this gene may facilitate DNA repair and muscle satellite cell regeneration [36]. Additionally, *GADD45A*’s role in cell cycle arrest [41] may contribute to its effects on endurance exercise. Meanwhile, *CXCL8* (C-X-C Motif Chemokine Ligand 8) expression was also significantly upregulated post-exercise. The protein encoded by this gene, IL-8, primarily mediates local inflammation-related immune responses [42], and its levels rise after endurance exercise [43], aligning with the findings of this study. This verifies that this gene participates in regulating cellular autophagy in the senescence pathway.

A significant increase was observed in *NFKBIA* (nuclear factor of kappa light polypeptide gene enhancer in B-cells inhibitor alpha) gene expression post-exercise, potentially related to inflammatory responses. It has been reported that heat shock can upregulate *NFKBIA* expression and hinder excessive inflammation-induced cell damage by inhibiting NF-κB activation [44]. Additionally, horses presented elevated body temperatures after endurance racing. Upregulated *NFKBIA* expression may protect cells during heat stress [45], retaining high-quality intracellular energy metabolism. Moreover, this study found that *MMP9* (Matrix metalloproteinase-9) expression significantly increased post-exercise. MMP9 is involved in cardiac repair and remodeling [46]. Serum MMP9 levels rise during endurance exercise [47], which is consistent with this study. Furthermore, *MMP9*-knockout mice exhibited a reduction in collagen accumulation [48], suggesting MMP9 may regulate endurance exercise through lipid metabolism and cardiac function modulation. *CXCL1* (C-X-C Motif Chemokine Ligand 1) expression was significantly upregulated after endurance racing, aligning with studies on human endurance exercise [49]. These genes were significantly enriched in the lipid metabolism/atherosclerosis pathway, indicating that their regulation is closely related to lipid metabolism and other factors.

In this study, the screened DMs were primarily enriched in pathways associated with lipid metabolism, including fatty acid biosynthesis, biosynthesis of unsaturated fatty acids, arachidonic acid metabolism, and one carbon pool by folate. The efficiency of lipid metabolism directly impacts the performance of horses in endurance events [50]. Fatty acid oxidation during exercise provides abundant ATP (cellular energy currency), offering a higher energy density compared to carbohydrates [17]. Consequently, the efficiency of fat metabolism is particularly crucial in prolonged exercise, helping athletes preserve stamina and endurance [14]. Furthermore, lipid metabolism contributes to adaptive changes in muscle cells. Exercise training enhances the number and function of mitochondria in muscles, boosting fatty acid oxidation capacity. This can heighten endurance while abating reliance on carbohydrates and lowering fatigue [51]. Studies have confirmed that lauric acid, myristic acid, palmitic acid, and stearic acid are major fatty acid substrates in aerobic metabolism, playing a pivotal role in athletic performance [52]. This study observed significant increases in these fatty acids post-endurance competition, consistent with findings on cross-country skiing [53]. This is primarily because, during prolonged exercise, the utilization of non-esterified fatty acids (NEFA) in working muscles is crucial for aerobic ATP synthesis [54]. Additionally, the content of major unsaturated fatty acids notably rises following high-intensity exercise [55]. In this study, the levels of arachidonic acid, docosahexaenoic acid, eicosapentaenoic acid, and linoleic acid were elevated post-endurance competition, conforming to previous research [53]. This may be attributed to changes in muscle uptake of these fatty acids [54]. The increased levels of these fatty acids denote that horses can diminish muscle glycogen breakdown more effectively during endurance events, thus lowering cardiac load [56] and sustaining high metabolic activity.

In summary, this study investigated the differences in mRNA and metabolites in the blood of Yili horses before and after an 80 km endurance race. The transcriptomics and metabolomics analysis results provide insights into the molecular characteristics of Yili horses’ blood surrounding endurance competitions. The findings demonstrate that the differential substances in the blood of Yili horses before and after endurance races are predominantly associated with lipid and amino acid metabolism, as the identified DEGs and DMs are mainly related to these two types of metabolism. A comprehensive analysis of these molecular changes may aid in determining the regulatory pathways of horses during endurance exercise and elucidating their underlying mechanisms. However, the specific roles of these DEGs and DMs in endurance racing remain obscure and warrant further exploration.

## 4. Materials and Methods

### 4.1. Animals and Sample Collection

This study included horses participating in the Yili horse 80 km series races. (These races are held weekly. Horses that win weekly races advance to the monthly races, those who win monthly races move on to the quarterly races, and winners of the quarterly races qualify for the annual finals.) Prior to racing, all horses underwent a veterinary examination to ensure they were in good health and free of lameness, metabolic disorders, or signs of fatigue. Blood samples were collected from all horses one day before the race (T0 group) and immediately after the race (within five minutes) (T1 group) using EDTA anticoagulant tubes. Post-race performance data for the competing horses were recorded. At the end of the year-round competition, blood samples from the top six horses, based on race results, were selected for transcriptomics and metabolomics analyses.

All blood samples were immediately pre-processed following collection. For metabolomics analysis, blood samples were centrifuged at 3000 rpm for ten minutes to derive plasma samples from the supernatant. For transcriptomics analysis, blood samples were mixed with three volumes of Trizol solution and gently shaken. All prepared samples were stored in liquid nitrogen until further use.

### 4.2. Total RNA Extraction and Library Construction

Total RNA was extracted from blood samples using the Spin Column Blood Total RNA Purification Kit (B518653, Sangon Biotech, Beijing, China) following the manufacturer’s instructions. RNA purity and concentration were assessed using a NanoDrop 2000 (Thermo Fisher Scientific Inc., Waltham, MA, USA) and Agilent 2100 Bioanalyzer (Agilent, Baden-Württemberg, Germany). RNA samples with an RIN (RNA Integrity Number) greater than 8.5 were used for further analysis. A mRNA-seq library was constructed and quantified initially using a Qubit 2.0 Fluorometer (Thermo Fisher Scientific Inc., Waltham, MA, USA). The effective concentration of the library was precisely determined using qRT-PCR. After confirming the library’s quality, Illumina sequencing was conducted. Library construction and transcriptome sequencing were fulfilled by Beijing Novogene Technology Co., Ltd. (Beijing, China).

### 4.3. RNA-Seq Data Processing and Analysis

Raw data (reads) were filtered to remove adaptor sequences, reads containing N, and low-quality reads (where more than 50% of bases had Qphred ≤ 5). Error rates and GC content distributions were checked to obtain valid data (clean reads). Hisat2 software (v2.2.1, Johns Hopkins University, Baltimore, MD, USA) was adopted to align clean reads with the horse genome data (Equus Caballus: EquCab3.0) from the NCBI Genome database and to annotate them, obtaining information on read locations in the reference genome. Gene expression levels were calculated using the FPKM method (Fragments Per Kilobase of exon model per Million mapped fragments). Differential expression analysis was performed using DESeq2 software (1.46.0, Bioconductor), with |log2(FoldChange)| ≥ 2 and padj ≤ 0.01 as the criteria for identifying differentially expressed genes (DEGs). GO and KEGG enrichment analyses of DEGs were conducted using clusterProfiler (version 3.8.1, Bioconductor). Significant enrichment pathways were defined by padj < 0.05.

### 4.4. Real-Time Quantitative PCR Validation

Four DEGs were randomly selected for validation using real-time quantitative PCR. The amplification program consisted of 40 cycles: 94 °C for 30 s, 94 °C for 5 s, 60 °C for 15 s, and 72 °C for 10 s. Specific primers were designed using Primer 6.0 (Premier, Stewart, Canada). The primer information for the housekeeping gene *GAPDH* and the target genes *CD14* (Cluster of Differentiation 14), *TRIB1* (tribbles-1), *S100A8* (S100 calcium-binding protein A8), and *LTF* (Lactotransferrin) is provided in Table 2. Relative expression levels for each sample were calculated using the 2^−ΔΔCt^ method.

### 4.5. Metabolite Extraction

100 μL of plasma was taken out and placed in an EP tube, and 400 μL of 80% methanol–water solution was added. The mixture was vortexed, incubated on ice for five minutes, and then centrifuged at 15,000× *g* for 20 min at 4 °C. The supernatant was collected and diluted with mass spectrometry-grade water to a final methanol content of 53%. After centrifugation, the supernatant was gathered for LC-MS analysis.

### 4.6. LC-MS Analysis

LC-MS analysis was conducted on the platform provided by Novogene (China). A Vanquish UHPLC system (Thermo Fisher, Darmstadt, Germany) was employed for chromatographic analysis. Chromatographic conditions: A Hypersil Gold column (100 × 2.1 mm, 1.9 μm, Thermo Fisher Scientific Inc., Waltham, MA, USA) was adopted as the chromatography column. In positive ion mode, mobile phase A was 0.1% formic acid in water, and mobile B was methanol; in negative ion mode, mobile phase A was 5 mM ammonium formate in water (pH 9.0), and mobile B was methanol. The injection volume was 2 μL, and the column temperature was set to 40 °C. The mobile phase gradient was as follows: 0–3 min at 2% B; 3–10 min at 85% B; 10–10.1 min at 0% B; 10.1–11 min at 2% B. The mass spectrometer used was the Q Exactive™ HF/Q Exactive™ HF-X (Thermo Fisher, Darmstadt, Germany).

Mass spectrometry conditions: Upon electrospray ionization, mass spectra signals were collected using both positive and negative ion scanning modes within a range of 100–1500 *m*/*z*. The sheath gas flow rate was 35 psi, the auxiliary gas flow rate was 10 L/min, the temperature of the ion transfer tube was 320 °C, the temperature of the auxiliary gas heater was 350 °C, and the spray voltage was 3.5 kV. MS/MS secondary scans were performed in data-dependent mode.

### 4.7. Metabolite Identification and Screening

Raw metabolomics data were imported into Compound Discoverer™ 3.3 software (Thermo Fisher, Darmstadt, Germany) to screen retention time and *m*/*z* ratio, followed by peak extraction and peak area quantification. Molecular formulas were predicted from molecular ion peaks and fragment ions and compared with mzCloud (https://www.mzcloud.org/, accessed on 26 December 2024), mzVault, and Masslist databases. The MS mass error was set to <5 ppm. Based on the expression levels of metabolites in different samples, Ropls (R-3.4.3 package, Bioconductor) and Scipy (Python-3.5.0, Bioconductor) were adopted for principal component analysis (PCA) and correlation heatmap analysis, evaluating intra-group sample similarity and differences. The identified metabolites were annotated using KEGG (https://www.genome.jp/kegg/pathway.html, accessed on 26 December 2024), HMDB (https://hmdb.ca/metabolites, accessed on 26 December 2024), and LIPIDMaps (http://www.lipidmaps.org/, accessed on 26 December 2024) databases.

T-tests (two-tailed) were employed to calculate *p*-values for metabolites in each group. Differential metabolites (DMs) were screened based on VIP > 1.0, FC > 1.2/FC < 0.833, and *p*-value < 0.05. The KEGG database was used to analyze the pathways of identified DMs.

### 4.8. Integrated Analysis of Transcriptomics and Metabolomics

Based on the KEGG enrichment results from the comparison of metabolomics and transcriptomics data, co-enriched pathways were identified using KEGG pathways as entries. The Pearson correlation coefficient (r^2^) and *p*-value between DEGs and DMs were calculated and a correlation heatmap was depicted. A significant correlation between DMs and DEGs was determined when *p* < 0.05 (labeled by * in the figure).

## 5. Conclusions

Through the application of transcriptomics and metabolomics, this study highlights changes in gene expression and metabolic profiles in the blood of Yili horses before and after endurance racing. This research provides valuable reference data for studies on gene expression and metabolic regulation in equine endurance. These findings can substantially contribute to optimizing horse endurance performance and formulating targeted training programs.

## Figures and Tables

**Figure 1 ijms-26-02426-f001:**
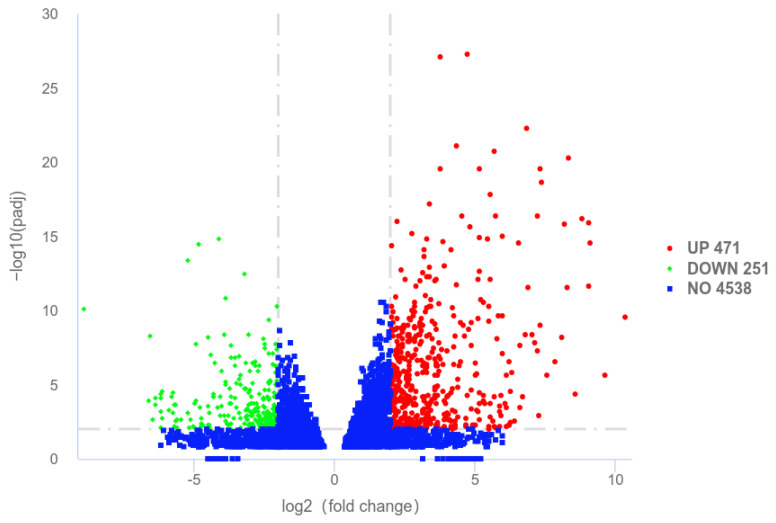
The volcano map of DEGs.

**Figure 2 ijms-26-02426-f002:**
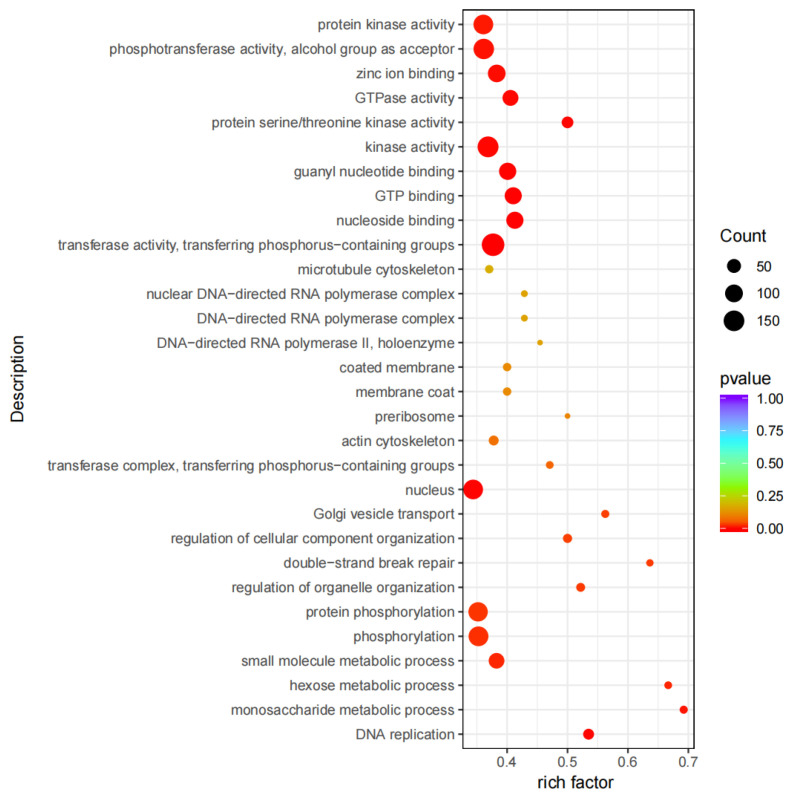
The scatter plot of GO enrichment analysis. The size of circle represents the count of genes annotated onto the GO Term.

**Figure 3 ijms-26-02426-f003:**
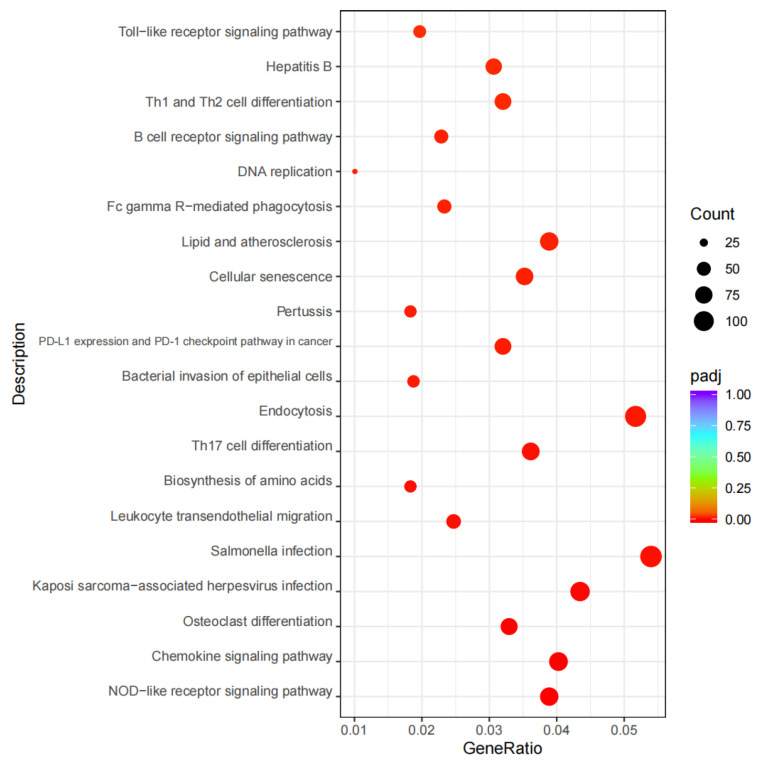
The scatter plot of KEGG enrichment analysis. The size of circle represents the count of genes annotated onto the KEGG Pathway.

**Figure 4 ijms-26-02426-f004:**
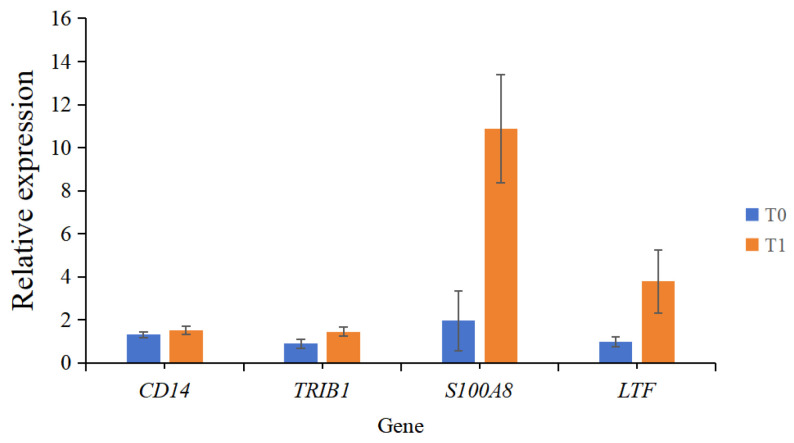
qRT-PCR verification of four random genes.

**Figure 5 ijms-26-02426-f005:**
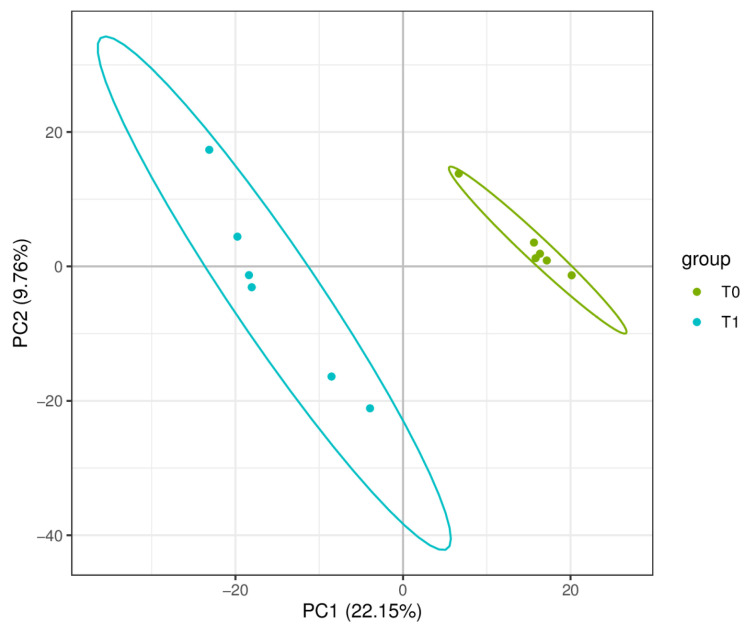
The Principal Component Analysis (PCA) of metabolites.

**Figure 6 ijms-26-02426-f006:**
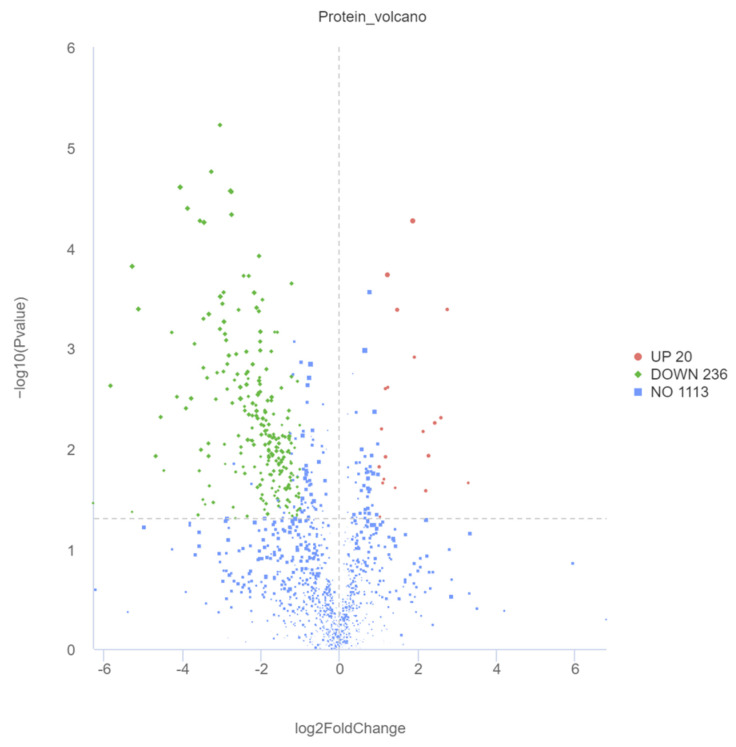
The volcano map of DMs.

**Figure 7 ijms-26-02426-f007:**
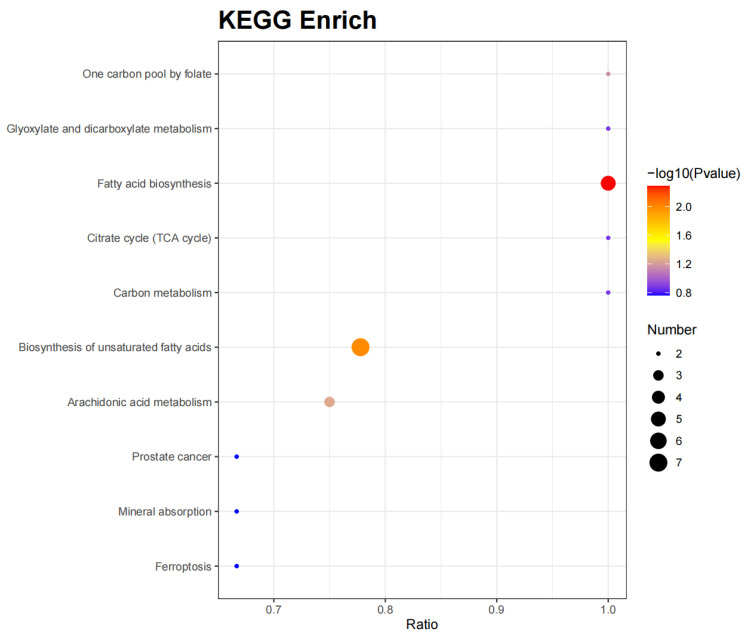
The bubble chart of the KEGG enrichment analysis.

**Figure 8 ijms-26-02426-f008:**
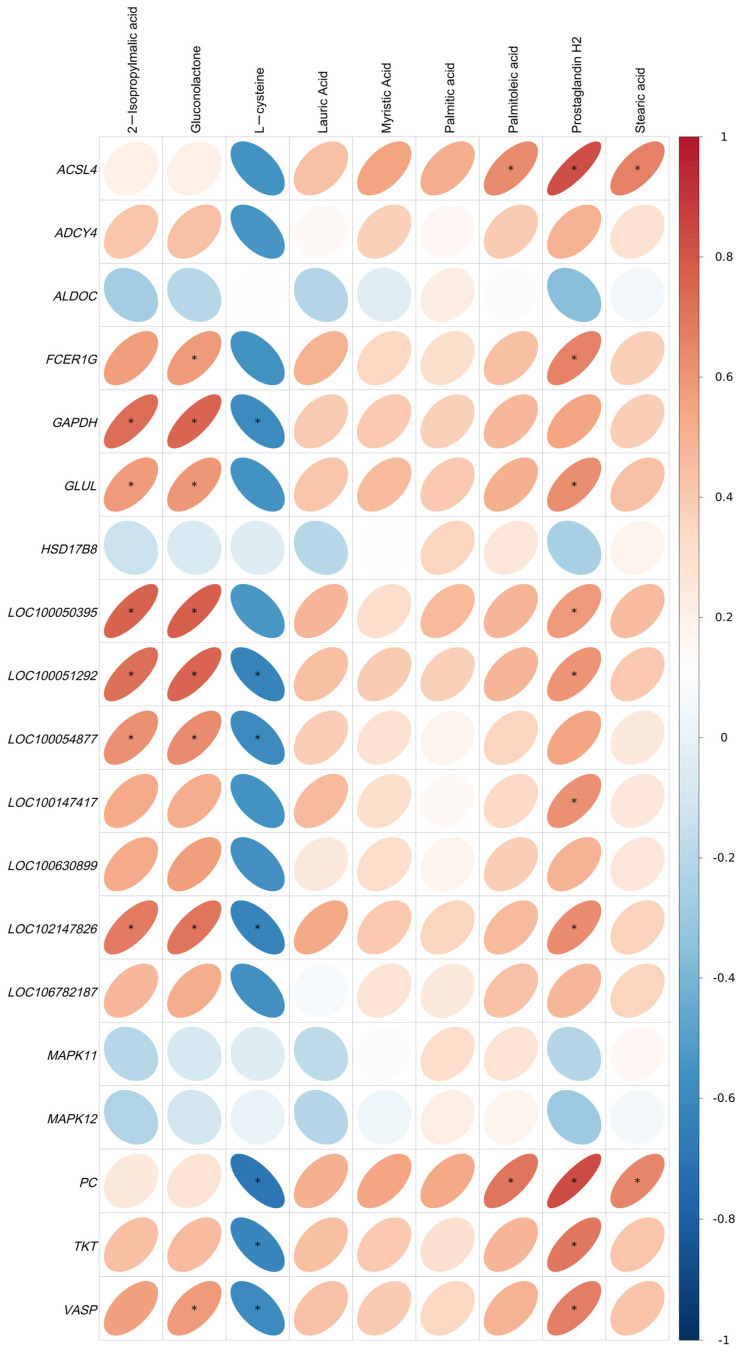
The correlation heatmap of DMs and DEGs. In the graph, blue indicates a negative correlation, and red indicates a positive correlation. The rounder the circle, the greater the Pearson correlation coefficient. * indicates *p* < 0.05.

**Table 1 ijms-26-02426-t001:** RNA-seq data quality check.

Group	Sample	Raw Date		Clean Date		Clean Ratio	Mapped Reads	Q20%	Q30%	GC Content%
		Read	Base (G)	Read	Base (G)					
T0	1	42,199,338	6.33	41,398,488	6.21	98.10	38,532,112 (93.08%)	98.05	94.62	56.36
	2	46,136,320	6.92	45,499,530	6.82	98.62	42,987,209 (94.48%)	98.20	94.92	55.25
	3	42,628,206	6.39	42,202,070	6.33	99.00	40,298,138 (95.49%)	98.03	94.42	54.3
	4	48,256,328	7.24	47,745,646	7.16	98.94	43,232,922 (90.55%)	98.03	94.55	58.74
	5	48,678,986	7.3	48,039,608	7.21	98.69	43,347,330 (90.23%)	97.82	94.10	60.34
	6	46,770,976	7.02	45,942,990	6.89	98.23	43,698,460 (95.11%)	98.10	94.64	53.21
T1	1	44,806,188	6.72	44,262,422	6.64	98.79	41,365,321 (93.45%)	98.04	94.55	56.69
	2	47,663,710	7.15	47,111,612	7.07	98.84	44,558,797 (94.58%)	98.08	94.67	57.95
	3	48,090,328	7.21	47,441,622	7.12	98.65	44,392,457 (93.57%)	97.85	94.21	58.2
	4	41,936,500	6.29	41,013,050	6.15	97.80	38,099,695 (92.9%)	97.91	94.40	57.29
	5	49,041,564	7.36	48,512,654	7.28	98.92	45,381,440 (93.55%)	97.84	94.16	56.55
	6	47,552,158	7.13	46,749,566	7.01	98.31	42,493,180 (90.9%)	98.09	94.78	57.69

**Table 2 ijms-26-02426-t002:** qRT-PCR primers.

Gene Name	Forward Primer Sequence (5′→3′)	Reverse Primer Sequence (5′→3′)	Size (bp)
*CD14*	GGAGCAGGTGCCTAAAGGACTA	AGTTCTGGTCTTGCTGCTTGGA	167
*TRIB1*	AAATCCGACGTGGACAGTTCTG	GCTTCCAAGACGGACTCAAAC	148
*S100A8*	GCCATCTATAGGGACGACTTGAA	GATGAGGAACTCCTGGAAGTTAACA	138
*LTF*	GATGGCGGTTTGGTGTATGA	GCTGCCCTTCTTCACTACGG	129
*GAPDH*	ATGGTGAAGGTCGGAGTAAACG	CATGGGTGGAATCATACTGAAACA	154

## Data Availability

The results of Transcriptomics sequencing of horse blood have been deposited to the BioProject with the accession number PRJNA1200936. Other data used and analyzed during the current study are available from the corresponding author upon reasonable request.

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
