# Peer review of "Regulatory Mechanisms of Yili Horses During an 80 km Race Based on Transcriptomics and Metabolomics Analyses"

_ijms, 2025, doi:10.3390/ijms26062426_

Round 1
Reviewer 1 Report
Comments and Suggestions for Authors
Using a transcriptomics and metabolomics approach, this manuscript investigates the physiological and molecular changes associated with equine endurance exercise. The well-designed study provides valuable insights into the regulatory mechanisms underlying endurance in horses. The integration of multi-omics data enhances our understanding of metabolic adaptations during prolonged exercise, making this research both relevant and significant.
Overall, the manuscript is well-written and presents an interesting and useful topic. My comments are below. They are minor and mainly focus on consistency, formatting, and clarity.
GENERAL COMMENT
When referring to genes, use italics; when referring to proteins, use regular font. For example, in Line 178: “The proteins encoded by GLUL are involved in glutamine synthesis. High GLUL activity generally facilitates the biosynthesis of glutamine [30], an amino acid essential for protein and energy metabolism.” Please write genes in italics also in Figure 4, Figure 8, and Table 2.
ABSTRACT
Line 12: The term "in vivo" is unnecessary.
Line 20: Did you mean "integrative" instead of "interactive"?
RESULTS
Line 82: "table" should be written as "Table" (capitalized).
Line 85: In Materials & Methods, it is written as padj, but here it appears as padj (italics for p). Please ensure consistency.
Figure 8. The gene names and metabolite labels in the figure are difficult to read. Please consider improving the font size and resolution to enhance readability.
Line 121: Instead of "PCA of 12 samples revealed a clear separation between the two groups (Figure 5).", it would be clearer to say " PCA of six samples per time point (T0 and T1) revealed a clear separation between the two groups (Figure 5)."
Lines 150-154: This content should be moved to the Figure 8 caption.
DISCUSSION
- Please write full gene names when first mentioned.
- In the Results section, DEGs are reported based on T0 vs. T1, but in the Discussion, individual genes (e.g., GLUL) are described in terms of T1 vs. T0 (opposite direction). While this is mathematically correct, it may be confusing to readers. Please ensure the comparison direction remains consistent throughout the manuscript or explicitly clarify the shift when discussing individual genes. Additionally, in the Real-Time Quantitative PCR Validation (Figure 4), please confirm whether the same T0 vs. T1 logic was applied. If not, indicate any differences to avoid confusion. To enhance clarity, consider consistently presenting gene expression comparisons as T1 vs. T0 or clearly stating the shift when interpreting findings.
MATERIALS AND METHODS
- Sections 4.3, 4.4, and 4.7: Missing references or company names for the software and tools used.
- Section 4.7: "CD software" should be written as Compound Discoverer™ Software.
Author Response
Dear Editors and Reviewers:
Thank you for your letter and for the reviewers’ comments concerning our manuscript entitled “Regulatory Mechanisms of Yili Horses During an 80 km Race based on Transcriptomics and Metabolomics Analyses” (ID: ijms-3497902). These comments are all valuable and very helpful for revising and improving our paper, as well as the important guiding significance to our research. We have studied the comments carefully and have made corrections accordingly. We hope the new version will meet the standards for publication in the International Journal of Molecular Sciences.
Reviewer #1:
Using a transcriptomics and metabolomics approach, this manuscript investigates the physiological and molecular changes associated with equine endurance exercise. The well-designed study provides valuable insights into the regulatory mechanisms underlying endurance in horses. The integration of multi-omics data enhances our understanding of metabolic adaptations during prolonged exercise, making this research both relevant and significant.
Overall, the manuscript is well-written and presents an interesting and useful topic. My comments are below. They are minor and mainly focus on consistency, formatting, and clarity.
1.GENERAL COMMENT
When referring to genes, use italics; when referring to proteins, use regular font. For example, in Line 178: “The proteins encoded by GLUL are involved in glutamine synthesis. High GLUL activity generally facilitates the biosynthesis of glutamine [30], an amino acid essential for protein and energy metabolism.” Please write genes in italics also in Figure 4, Figure 8, and Table 2.
Reply: We agree with the reviewer's suggestions and have revised the Font for gene names.
2.Line 12: The term "in vivo" is unnecessary.
Reply:We agree with the reviewer's suggestions and removed “in vivo”. (line 12, page 1)
3.Line 20: Did you mean "integrative" instead of "interactive"?
Reply: We agree with the reviewer's suggestions and changed “interactive” to “integrative”. (line 21, page 1)
4.Line 82: "table" should be written as "Table" (capitalized).
Reply: We agree with the reviewer's suggestions and revised it. (line 86, page 2)
- Line 85: In Materials & Methods, it is written as padj, but here it appears as padj (italics for p). Please ensure consistency.
Reply: We agree with the reviewer's suggestions and revised it. (line 89, page 3)
- Figure 8. The gene names and metabolite labels in the figure are difficult to read. Please consider improving the font size and resolution to enhance readability.
Reply: We agree with the reviewer's suggestions and increased Font sizes. (line 160, page 8)
7.Line 121: Instead of "PCA of 12 samples revealed a clear separation between the two groups (Figure 5).", it would be clearer to say " PCA of six samples per time point (T0 and T1) revealed a clear separation between the two groups (Figure 5)."
Reply: We agree with the reviewer's suggestions and revised it. (line 126-127, page 5)
8.Lines 150-154: This content should be moved to the Figure 8 caption.
Reply: We agree with the reviewer's suggestions and moved it. (line 162-167, page 8)
9.Please write full gene names when first mentioned.
Reply: We agree with the reviewer's suggestions and revised it.
10.In the Results section, DEGs are reported based on T0 vs. T1, but in the Discussion, individual genes (e.g., GLUL) are described in terms of T1 vs. T0 (opposite direction). While this is mathematically correct, it may be confusing to readers. Please ensure the comparison direction remains consistent throughout the manuscript or explicitly clarify the shift when discussing individual genes. Additionally, in the Real-Time Quantitative PCR Validation (Figure 4), please confirm whether the same T0 vs. T1 logic was applied. If not, indicate any differences to avoid confusion. To enhance clarity, consider consistently presenting gene expression comparisons as T1 vs. T0 or clearly stating the shift when interpreting findings.
Reply: We agree with the reviewer's suggestions and revised it. The updated version now reads as follows:
Compared to the T0 group, the T1 group had 471 DEGs significantly upregulated and 251 ones significantly downregulated. (line 90-91, page 3)
and the expression levels of four genes in T1 group were higher than those in T0 group, (line 120-121, page 5)
11.MATERIALS AND METHODS
Sections 4.3, 4.4, and 4.7: Missing references or company names for the software and tools used.
Reply: We agree with the reviewer's suggestions and revised it.
12.Section 4.7: "CD software" should be written as Compound Discoverer™ Software.
Reply: We agree with the reviewer's suggestions and revised it. (line 348, page 12)
The modified sections in the manuscript have been highlighted in red.
Thank you very much for your attention and time. Look forward to hearing from you.
Yours sincerely,
MENG Jun, YAO Xinkui
February 26, 2025
College of Animal Science, Xinjiang Agricultural University
Reviewer 2 Report
Comments and Suggestions for Authors
Dear Authors,
Your research highlighting differentially expressed genes (DEGs) and differential metabolites (DMs) of YILI horses involved in physiological changes in metabolism and molecular pathways responsible for maintaining in vivo balance during endurance racing is a well-documented work, as a result of complex work. I congratulate you for this work.
The research is well-conducted. Each section is correctly presented both in terms of content and editing. I particularly noted the discussion section, which is complex and offers pertinent explanations of the functional changes that occur under conditions of exposure of the organism (animal or human) to intense physical exercise.
The research is important both from a theoretical point of view, by identifying DEGs and DMs involved in the changes that occur in the metabolic profile of horses subjected to endurance racing, but also practically, by the fact that it can constitute a starting point in optimizing horse endurance performance and formulating targeted training programs.
As recommendations I have a few aspects:
Harmonization of the number of horses specified in the abstract (five) with that in the material and method (blood samples from the top six horses).
Reporting animal research in adherence with the ARRIVE guidelines 2.0 ensures transparent and thorough reporting. Specify if your study is in compliance with these guidelines.
Author Response
Dear Editors and Reviewers:
Thank you for your letter and for the reviewers’ comments concerning our manuscript entitled “Regulatory Mechanisms of Yili Horses During an 80 km Race based on Transcriptomics and Metabolomics Analyses” (ID: ijms-3497902). These comments are all valuable and very helpful for revising and improving our paper, as well as the important guiding significance to our research. We have studied the comments carefully and have made corrections accordingly. We hope the new version will meet the standards for publication in the International Journal of Molecular Sciences.
Reviewer #2:
Your research highlighting differentially expressed genes (DEGs) and differential metabolites (DMs) of YILI horses involved in physiological changes in metabolism and molecular pathways responsible for maintaining in vivo balance during endurance racing is a well-documented work, as a result of complex work. I congratulate you for this work.
The research is well-conducted. Each section is correctly presented both in terms of content and editing. I particularly noted the discussion section, which is complex and offers pertinent explanations of the functional changes that occur under conditions of exposure of the organism (animal or human) to intense physical exercise.
The research is important both from a theoretical point of view, by identifying DEGs and DMs involved in the changes that occur in the metabolic profile of horses subjected to endurance racing, but also practically, by the fact that it can constitute a starting point in optimizing horse endurance performance and formulating targeted training programs.
As recommendations I have a few aspects:
1.Harmonization of the number of horses specified in the abstract (five) with that in the material and method (blood samples from the top six horses).
Reply:Indeed, this oversight is attributable to us. We are very sorry for this error. By checking the study data, we found that we took blood samples from six horses, so we revised the sample size in the abstract section. We sincerely apologize for this mistake. The updated version now reads as follows:
This study collected blood samples from six Yili horses before and after an 80 km race and conducted transcriptomics and metabolomics analyses, yielding 722 DEGs and 256 DMs. (line 16, page 1)
2.Reporting animal research in adherence with the ARRIVE guidelines 2.0 ensures transparent and thorough reporting. Specify if your study is in compliance with these guidelines.
Reply: In response to the reviewers' comments, we have revised the manuscript to ensure full compliance with the ARRIVE guidelines 2.0 by implementing all required reporting elements.
The modified sections in the manuscript have been highlighted in red.
Thank you very much for your attention and time. Look forward to hearing from you.
Yours sincerely,
MENG Jun, YAO Xinkui
February 26, 2025
College of Animal Science, Xinjiang Agricultural University